# Adrenergic-mediated loss of splenic marginal zone B cells contributes to infection susceptibility after stroke

Laura McCulloch[1], Craig J. Smith[2,3] & Barry W. McColl[1]

Infection is a major complication of acute stroke and causes increased mortality and morbidity; however, current interventions do not prevent infection and improve clinical outcome in stroke patients. The mechanisms that underlie susceptibility to infection in these patients are unclear. Splenic marginal zone (MZ) B cells are innate-like lymphocytes that provide early defence against bacterial infection. Here we show experimental stroke in mice induces a marked loss of MZ B cells, deficiencies in capturing blood-borne antigen and suppression of circulating IgM. These deficits are accompanied by spontaneous bacterial lung infection. IgM levels are similarly suppressed in stroke patients. β-adrenergic receptor antagonism after experimental stroke prevents loss of splenic MZ B cells, preserves IgM levels, and reduces bacterial burden. These findings suggest that adrenergic-mediated loss of MZ B cells contributes to the infection-prone state after stroke and identify systemic B-cell disruption as a target for therapeutic manipulation.

[1] The Roslin Institute and R(D)SVS, University of Edinburgh, Easter Bush, Midlothian EH25 9RG, UK. [2] Stroke and Vascular Research Centre, University of Manchester, Manchester Academic Health Science Centre, Manchester M6 8HD, UK. [3] Greater Manchester Comprehensive Stroke Centre, Department of Medical Neurosciences, Salford Royal NHS Foundation Trust, Salford M6 8HD, UK. Correspondence and requests for materials should be addressed to B.W.M. (email: barry.mccoll@roslin.ed.ac.uk).

Clinical outcome in stroke patients is influenced not only by the primary brain injury but also neurological and medical complications. Infection is the most common complication of stroke, affecting up to one third of patients, and is independently associated with increased short-term and long-term mortality and morbidity[1,2]. Infections of bacterial origin affecting the respiratory or urinary tracts are the most prevalent and pneumonia itself is associated with a threefold increase in mortality and poorer functional outcome in survivors[1,3,4]. In addition to dysphagia and immobility, stroke-induced impairments in some aspects of systemic immunity are thought to contribute to risk of infection after stroke[2,5–7]. A general reduction in cellularity of systemic lymphoid tissues and blood has been described in experimental models of stroke and in patients[2,8]. Impaired function of various T-cell subsets has also been reported after experimental stroke in mice and is associated with spontaneous pneumonia[2,8–10]. Activation of autonomic neural pathways seems to be central to these systemic immune alterations[2,11,12].

Infection occurs most frequently in the first few days after stroke; therefore, deficits in conventional mechanisms of adaptive immunity, which are slowly activated, are unlikely to account for the initial susceptibility to infection[1,3,4]. There is increasing awareness of the importance of lymphocytes with innate-like functions in tissue homeostasis, immune regulation and infection control[13]. Marginal zone (MZ) B cells are a subset of innate-like lymphocytes in the MZ of the spleen, an important interface between the circulation and the immune system. MZ B cells mediate rapid responses to bacterial infection within 1–3 days after pathogen encounter by rapidly producing polyreactive immunoglobulin M (IgM) antibodies that recognise highly conserved microbial molecular patterns. This response is a crucial early anti-bacterial defence mechanism thought to bridge the temporal gap until conventional follicular B cells can respond in a T-cell-dependent manner[14–19]. Individuals who lack spleens due to congenital dysfunction or surgery, or have disruptions to their splenic MZ, are susceptible to similar strains of encapsulated bacteria that typically cause lung infections in stroke patients[20]. The susceptibility to these infections in asplenic individuals is generally attributed to a lack of MZ B-cell-derived, T-cell-independent, IgM and IgG antibody specific for bacterial capsular polysaccharides[21–23]. MZ B cells are one of the major cellular sources of IgM produced early after infection and individuals with IgM deficiency are also at particular risk of bacterial respiratory infection[24–26]. Thus there is an important functional relationship among innate-like functions of splenic MZ B cells, IgM and the lung that is essential for anti-bacterial defence. It is therefore pertinent to identify if innate-like B-cell anti-bacterial defences are affected by stroke, a phenomenon that has not been investigated previously in the context of any central nervous system (CNS) injury. In general support that systemic B-cell populations may be sensitive to CNS injury, a loss of B-cell populations and reduced antibody production by follicular B cells in response to immunisation with T-cell-dependent antigen was shown in experimental models of spinal cord trauma, although assessment was several weeks after injury[27–29]. Therefore, given that most infections occur in the first few days after stroke, and the established role of innate-like B cells in rapid anti-bacterial defence against strains typically affecting patients, we sought to identify if systemic innate-like B-cell functions are affected by ischaemic stroke and contribute to infection susceptibility.

We show that experimental stroke in mice causes rapid loss of MZ B cells associated with impaired IgM production and spontaneous bacterial infection. We also demonstrate lower concentrations of circulating IgM in patients with acute ischaemic stroke and that IgM levels are most suppressed in patients who develop infection. Adrenergic signalling mediates these deficits, suggesting involvement of autonomic pathways in brain-immune communication affecting B-cell function after stroke. Blockade of adrenergic signalling after experimental stroke using propranolol prevents loss of MZ B cells, restores circulating IgM levels and reduces infection. These data reveal loss of innate-like B-cell populations and their associated functions as an important mechanism contributing to infection susceptibility after stroke and highlight this pathway as a potential target for intervention.

## Results

**Disruption to splenic B-cells after experimental stroke.** The microanatomical organisation of splenic immune cells is of crucial importance to B-cell anti-microbial function, for example, by enabling appropriate exposure to antigen and interactions with other cell types in the optimal locations[30–32]. We first investigated gross changes to splenic B-cell populations in relation to general structural and cellular alterations after experimental stroke in mice induced by transient (40 min) occlusion of the middle cerebral artery occlusion (MCAO). This MCAO model produces moderately large brain infarction affecting cortical and subcortical tissue (see Supplementary Fig. 1). MCAO resulted in marked disruption to the splenic microarchitecture with a loss of lymphocyte segregation and B-cell follicular structures which was consistent throughout the spleen (Fig. 1a). An extensive loss of mature (B220[+]) B cells in the white pulp (WP) occurred as early as 1 day after MCAO and was sustained to 7 day (Fig. 1b,c). A decrease in the area of CD3[+] T-cell immunoreactivity was also detected (Fig. 1b,d). CD21[+] follicular dendritic cell (FDC) networks, which are normally found in B-cell follicles and are dependent on B-cell-derived signals for maturation and maintenance, were condensed and reduced in area (Fig. 1c,f). In contrast, the number of F4/80[+] macrophages within the WP was increased in mice 1–7 days after MCAO in comparison to naïve and sham controls (Fig. 1e,g). As expected and consistent with the correlation between stroke severity and immunosuppression in patients, modifying the extent of brain damage influenced splenic B-cell disruption. 20 min MCAO generated small infarcts restricted to striatal tissue (Supplementary Fig. 1) and resulted in negligible splenic disruption whereas 40 min (as above) or 60 min MCAO produced moderate to large cortical/subcortical infarcts (Supplementary Fig. 1) and caused severe splenic disturbance. Consistent with previous studies, spleen weights were reduced after MCAO and inversely correlated with infarct volume (Fig. 1h). Immunolabelling of B cells to distinguish WP showed that larger infarcts were associated with reduced area of WP (Fig. 1i). These data demonstrate marked disruption to B-cell population size and positioning after MCAO that occurs in the context of gross disorganisation of the entire splenic WP.

**Experimental stroke induces a loss of marginal zone B cells.** Although we have demonstrated multiple splenic immune cell populations are affected by experimental stroke, innate-like splenic MZ B cells were of particular interest because their critical importance to immune surveillance and rapid anti-bacterial response is highly relevant to the incidence of infection in the first few days after stroke[30]. MZ B cells express high levels of CD1d, IgM and CD21/35 (also expressed on FDC within the B-cell follicle). Marked loss of immunoreactivity for all these markers confirmed a substantial loss of MZ B cells as early as 1 day after MCAO and remained significantly reduced up to 5 day later (Fig. 2a–c). MZ B cells in mice do not recirculate in the blood but can shuttle to and from the B-cell follicle. Negligible MZ B-cell-specific immunolabelling could be detected within the

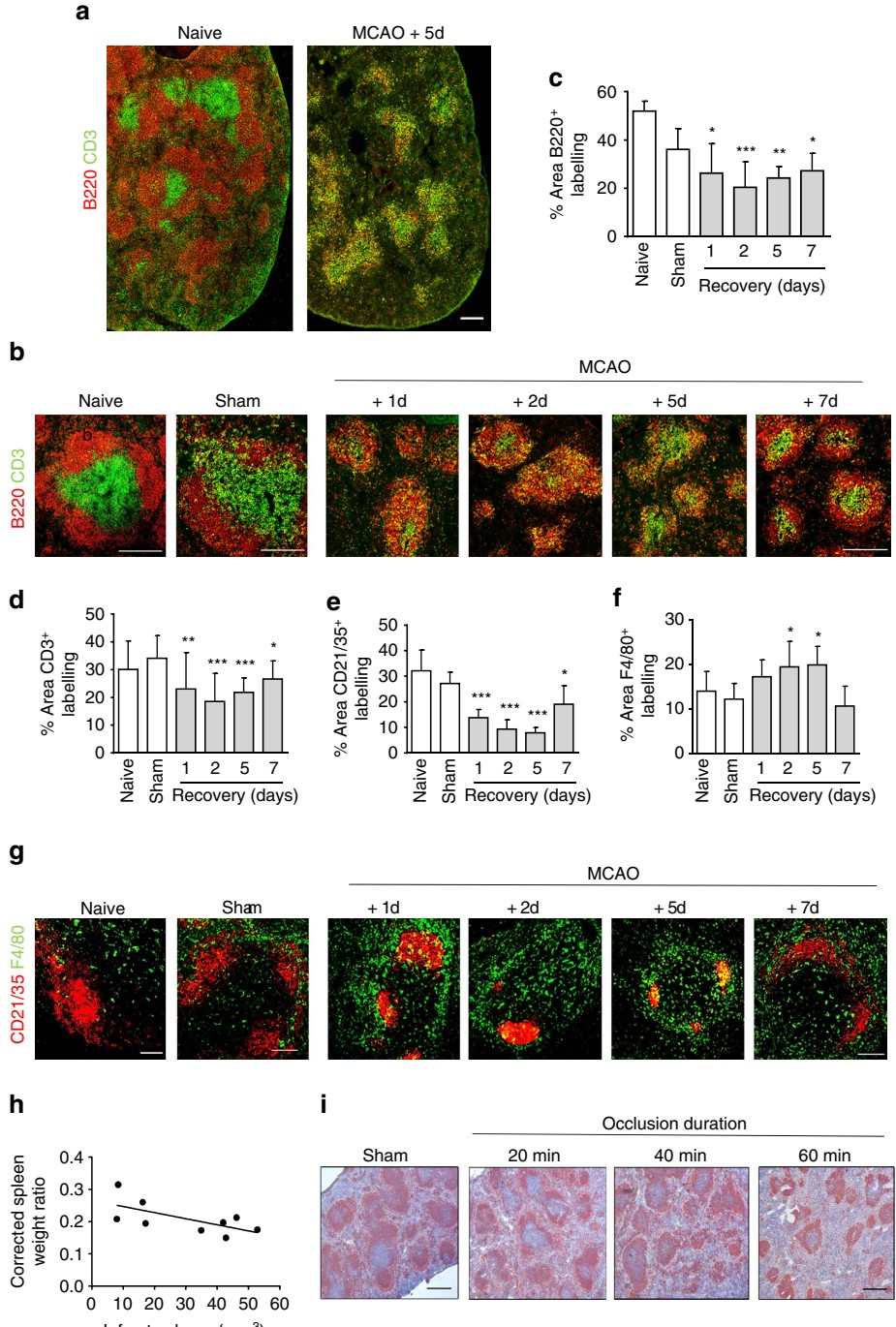

**Figure 1 | Cellular loss and WP disorganisation after experimental stroke.** (**a**) Tiled confocal images of splenic B (red, B220) and T-cell (green, CD3) immunolabelling show a reduction in WP area and disruption to remaining cellular localization. (**b**) Immunofluorescent labelling of splenic B (red, B220) and T cells (green, CD3) in naïve mice, mice recovered 1–7 days after MCAO or sham-operated mice. (**c**) Quantification of the percentage area of B220$^+$ immunolabelling per 846.28 µm$^2$ image in spleens from naïve mice or mice recovered from sham or MCAO surgery confirm the area of B-cell labelling is significantly reduced 1–7 days after MCAO in comparison to sham-operated controls. (**d**) Area of CD3$^+$ T-cell labelling is also reduced. (**g**) Immunolabelling of CD21/35$^+$ MZ B cells and FDC (red) and F4/80$^+$ macrophages (green) in spleens from naïve mice, mice recovered 1–7 days after MCAO or sham-operated mice show increased macrophages and a reduced area of FDC networks 1–7 days after MCAO in comparison to sham-operated controls. (**e**) Quantification of CD21/35$^+$ FDC and B-cell immunolabelling confirm a reduced area of labelling 1–7 days after MCAO whereas (**f**) there is an increase in the area of F4/80$^+$ macrophage labelling (naïve $n = 4$, sham $n = 4$, 1d $n = 4$, 2d $n = 9$, 5d $n = 5$, 7d $n = 7$). (**h**) Spleen weight, corrected for total body weight, inversely correlates with the infarct volume ($n = 9$). (**i**) Immunolabelling of B cells (red, B220$^+$) in spleens recovered for 2 days from occlusions of 20 min, 40 min, 60 min or sham surgery demonstrate the extent of disruption to splenic lymphoid tissue is dependent on the extent of injury in the brain. WP area and B-cell follicular structure is preserved in mice after sham surgery or 20 min MCAO. However 40 or 60 min MCAO results in loss of WP and rings of B cells forming within the WP. (Sham $n = 4$, 20 min $n = 7$, 40 min $n = 7$, 60 min $n = 7$). Scale bars (**a,b**) 200 µm; (**g**) 100 µm; (**i**) 500 µm. Data show mean + s.d.; *$P < 0.05$; **$P < 0.01$; ***$P < 0.001$; (**c–f**) one-way ANOVA with Bonferroni correction comparing all MCAO recovery time points to sham.

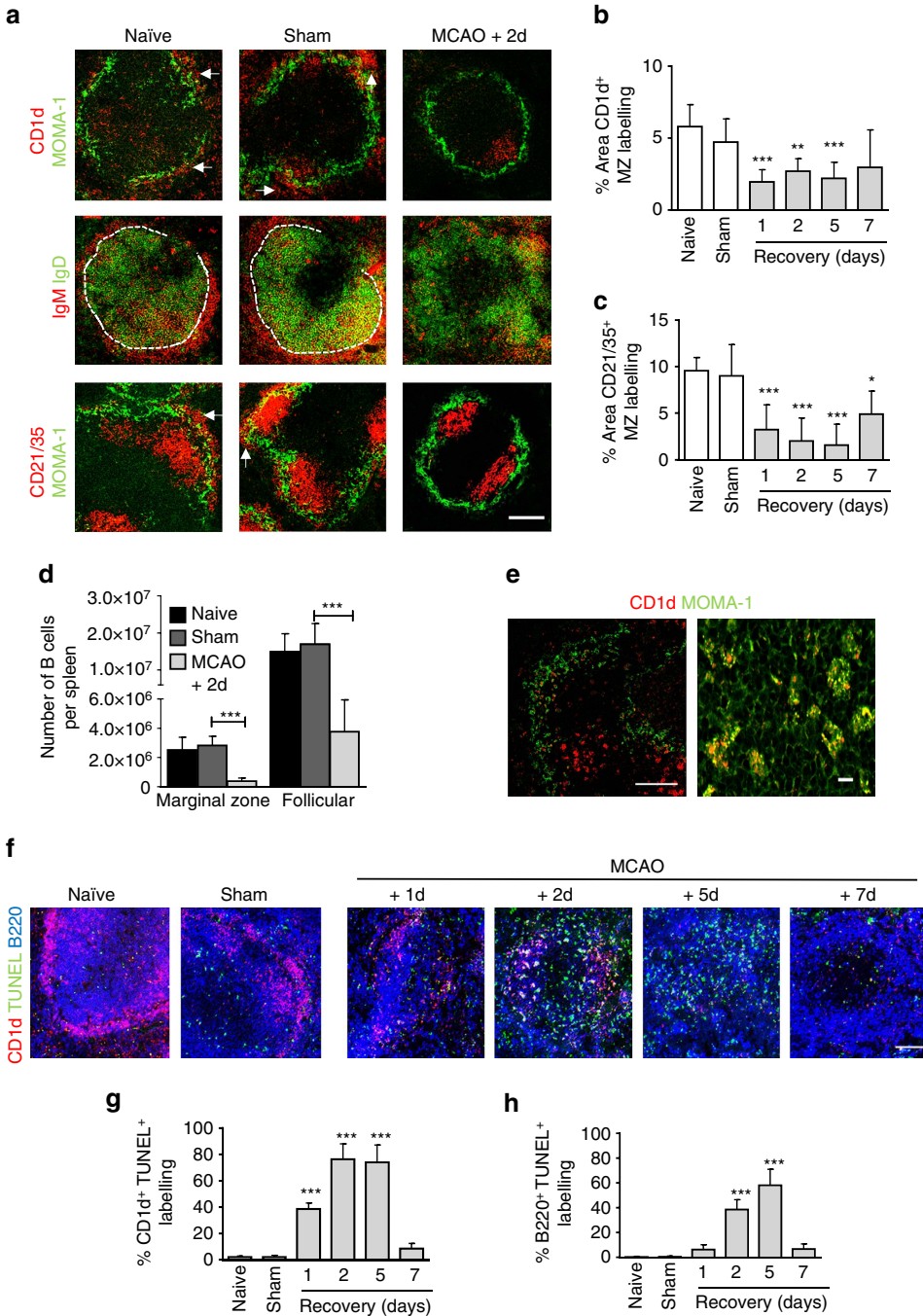

**Figure 2 | Experimental stroke induces a loss of MZ B cells.** (**a**) Immunolabelling of MZM macrophages (green, MOMA-1) with MZ B-cell markers (red, CD1d, and CD21/35) demonstrate the absence of MZ B cells (white arrows) from the MZ 2 days after MCAO in comparison to sham-operated controls. Immunolabelling to distinguish MZ (red, IgM) from follicular B cells (green, IgD) confirms this. Quantification of MZ B-cell immunolabelling outside the MOMA-1$^+$ MZM macrophage border using CD1d (**b**) or CD21/35 (**c**) confirm significant reductions in MZ B-cell labelling after MCAO in comparison to sham-operated controls (naïve $n = 4$, sham $n = 4$, 1d $n = 4$, 2d $n = 9$, 5d $n = 5$, 7d $n = 7$). CD23 and CD21/35 expression on cells gated for CD45R$^+$ labelling from a constant number of gated total events was used to distinguish MZ (B220$^+$CD23$^-$CD21/35$^{hi}$) from follicular (B220$^+$CD23$^+$CD21/35$^{int}$) B cells (Supplementary Fig. 1e). (**d**) Absolute counts of splenic MZ and follicular B cells using flow cytometry show both are significantly reduced 2 days after MCAO in comparison to sham-operated controls (naïve $n = 4$, sham $n = 4$, MCAO $n = 6$). (**e**) MZ B-cell immunolabelling (CD1d; red) can be seen within MZ metallophillic macrophages (green, MOMA-1) 2 days after MCAO. (**f**) TUNEL (green) staining, a marker of cell death, alongside B cells (blue, B220) and MZ B cells (red, CD1d) shows increased death of follicular and MZ B cells 1–7 days after MCAO in comparison to sham-operated controls. (**g**) Quantitative analysis of the percentage of CD1d$^+$ labelling that co-localises with TUNEL, as a measure of apoptotic MZ B cells, show increased TUNEL$^+$ CD1d MZ B cells 1–5 days after MCAO in comparison to sham-operated controls. (**h**) Similar analysis of CD1d$^-$ B220$^+$ labelling that co-localises with TUNEL, as a measure of apoptotic follicular B cells, shows a significant increase in TUNEL$^+$ follicular B cells 2–5 days after MCAO in comparison to sham-operated controls (naïve $n = 4$, sham $n = 4$, 1d $n = 4$, 2d $n = 9$, 5d $n = 5$, 7d $n = 7$). Scale bars (**a**,**f**) 200 μm (**e**) left panel 100 μm, right panel 10 μm. Data show mean + s.d.; *$P < 0.05$; **$P < 0.01$; ***$P < 0.001$; (**b**,**c**,**g**,**h**) one-way ANOVA with Bonferroni correction comparing all MCAO recovery time points to sham, (**d**) unpaired $t$-test.

follicle and was mostly associated with macrophages (Fig. 2a,e). Quantitative analysis by flow cytometry showed that total splenocyte counts were reduced by 65% 2 day after MCAO compared to sham-operated animals (Supplementary Fig. 2a). Consistent with the above data, only 13% of B220[+] CD23[−]CD21/35[hi] MZ B cells and 25% of B220[+]CD23[+] CD21/35[+] follicular B cells remained 2 day after MCAO in comparison to sham-operated controls (Fig. 2d, gating strategy presented in Supplementary Fig. 2b).

Several mechanisms could contribute to this loss of MZ B cells, including death, migration or lack of repopulation (see also below). Increased TUNEL labelling was present in spleens 1–7 days after MCAO in comparison to sham and naïve controls (Fig. 2e). Furthermore, a large percentage of MZ B cell (Fig. 2f) and follicular B cell (Fig. 2g) immunolabelling co-localizing with TUNEL was evident after MCAO even at time points when few MZ B cells were present. Labelling of MZ B-cell marker (CD1d) was detected within MOMA-1 (CD169)[+] MZ macrophages (Fig. 2h). Together, these data show that stroke induces a substantial loss of splenic B cells, notably MZ B cells, most likely through death and removal by local phagocytes rather than relocation to the B-cell follicle.

**Splenic transcriptome alterations after experimental stroke**. To gain insight to molecular pathways that may contribute to and be affected by MCAO-induced splenic B-cell loss, we performed transcriptomic analysis on spleens at the peak of disruption to lymphoid tissue microarchitecture (5 day after MCAO, see Fig. 1b). Comparison of splenic transcriptomes in MCAO- and sham-operated mice showed differential expression of 4,396 transcripts including 1,818 transcripts down-regulated and 2,578 transcripts up-regulated after MCAO ($P < 0.05$, fold change $\geq 1.5$, Fig. 3a). Markedly down-regulated genes included those highly expressed by B cells and essential to B-cell function including *Cr2*, involved in the capture of opsonised antigen by MZ B cells, *Cxcr5*, important in maintenance of B cells in the follicle and *Cd40*, important in B-cell activation (Fig. 3b). Validation of microarray data at the RNA and protein level was carried out on selected mediators by quantitative PCR (qPCR) and immuno-histochemistry (Supplementary Fig. 3a–c). The Gene Ontology biological processes B-cell activation, lymphocyte activation and immune system development were significantly over-represented in the set of MCAO down-regulated genes (Fig. 3c) as were the Kyoto Encyclopedia of Genes and Genomes (KEGG) pathways haematopoietic cell lineage, primary immunodeficiency and B-cell receptor signalling (Supplementary Fig. 3d). Within the KEGG Hematopoietic cell lineage pathway, decreased expression of many genes involved in mature B-cell development was particularly apparent (Fig. 3d). B cells are reliant on factors produced by resident stromal and immune cell populations for their maintenance, maturation and retention in the MZ or follicle. In contrast to the loss of cell intrinsic receptors on B cells that are crucial for their survival (for example, BAFFR (*Tnfrsf13c*); Fig. 3e) or positioning (for example, LFA-1 (*Itgal* and *Itgb2*); Supplementary Fig. 3e), expression of stromal cell-derived ligands that promote B-cell survival (for example, BAFF (*Tnfsf13b*); Fig. 3e) and cellular adhesion molecules that mediate B-cell localization (for example, *Vcam1*; Supplementary Fig. 3f) remained unchanged or increased. These data suggest the effects of MCAO are sufficient to overcome homeostatic mechanisms that regulate B-cell survival, maintenance and positioning within the spleen.

**Experimental stroke disrupts MZ anti-bacterial function**. The splenic MZ is optimised for the screening and capture of blood-borne pathogens that trigger innate-like rapid antibody responses

by MZ B cells and the trafficking of antigen to the B-cell follicle where development of slower adaptive immune responses take place. Given the marked loss of MZ B cells induced by MCAO, we next determined if these alterations led to impairments in key MZ B-cell-dependent functions. At 2 day after MCAO we assessed the ability of MZ B cells to capture model antigen PE-anti-CD21 from the circulation. In the spleens of naïve or sham-operated animals, PE-anti-CD21 was detected throughout the MZ 1 h after intravenous (i.v.) administration and co-localized with the MZ B-cell marker CD21/35 indicating efficient capture. In contrast, there was almost a complete absence of PE-anti-CD21 detected in the MZ after MCAO (Fig. 4a,c).

MZ B cells can regulate other cellular populations within the MZ, including MZ macrophages, which also capture blood-borne antigen. MZ B cells were previously shown as essential for MZ macrophage expression of the c-type lectin SIGN-R1 (ref. 33). SIGN-R1 recognises bacterial capsular polysaccharide and its expression is important for protection against pneumococcal infection[34,35]. No deficits in SIGN-R1 expression by MARCO[+] MZ macrophages were detected after MCAO despite the depletion of the MZ B cells described above (Fig. 4b). Furthermore, MZ macrophage-targeted model antigen, dextran-FITC, was efficiently captured by MARCO[+] MZ macrophages in naïve, sham and MCAO-operated animals 1 h after injection (Supplementary Fig. 4) and was efficiently trafficked to the FDC network by 24 h after i.v. administration (Fig. 4d,e). These data demonstrate a loss of splenic MZ antigen capture selectively affecting MZ B cells after MCAO.

Unstimulated MZ B cells work in synergy with splenic B1 B cells to contribute to natural IgM in the circulation under normal physiological conditions and within 24 h of stimulation give rise to plasma MZ B cells secreting high levels of IgM. This early T-independent antibody response is critical for early defence against many bacterial and viral pathogens[16]. Circulating IgM concentration was significantly lower after MCAO compared to sham-operated controls (Fig. 4f). Previous studies have reported that experimental stroke can cause spontaneous pneumonia and bacteraemia[2]. In agreement, we detected high bacterial counts in the lungs (Fig. 4g,i) and blood (Fig. 4h,j) of animals 1–7 days after MCAO that were not present in sham-operated controls suggesting opportunistic commensals as the most likely origin. Together these data suggest that MZ B-cell loss after MCAO causes impairments in immune functions critical for early anti-bacterial defence, notably antigen capture and IgM production, and that these are associated with spontaneous bacterial infection.

**Suppressed circulating IgM concentrations in stroke patients**. To determine if innate-like B-cell functions could be affected in patients after acute ischaemic stroke, IgM concentration was measured in plasma samples from patients and paired controls matched for age, sex and degree of atherosclerosis[36]. Plasma IgM concentration was significantly lower in stroke patients at admission (up to 12 h after onset of symptoms), 24 h and 5–7 days after stroke in comparison to controls (Fig. 5a). This profile correlates with the extensive loss of MZ B cells and reduction in circulating IgM observed at similar time points after experimental stroke above. Infection in patients was diagnosed using available clinical, radiological and laboratory data[36]. Three patients (3/37) reported infection preceding stroke and were excluded from subsequent analyses. Around a third (12/34) of patients were diagnosed with infection in the 14 day following stroke, including seven patients with pneumonia, which is similar to the incidence described previously[1,37]. The majority of these infections (8/12) occurred within the first 6 days after stroke onset. Minimum plasma IgM concentration measured up to 7 day after stroke was

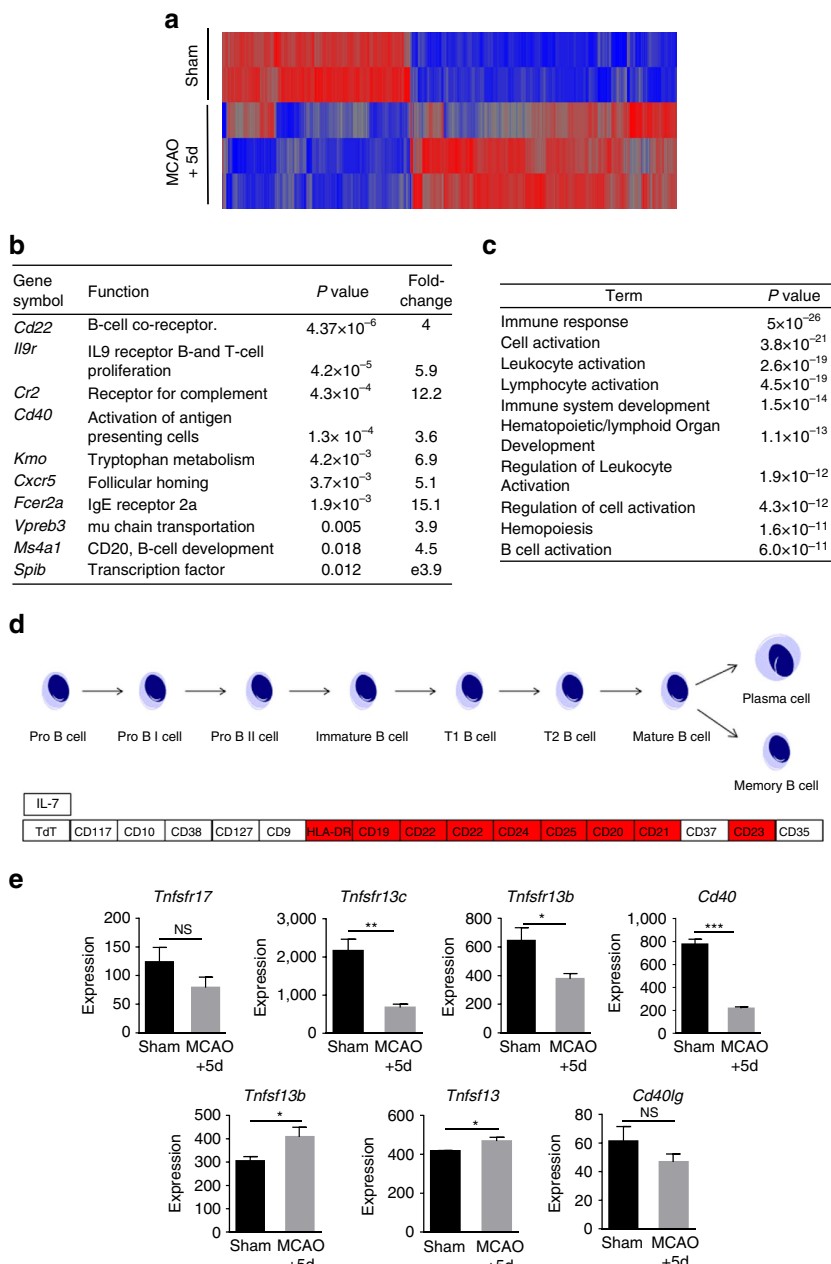

**Figure 3 | Genome-wide transcriptional changes in the spleen after experimental stroke.** (**a**) Heat map demonstrating the expression profile of 4396 transcripts differentially expressed in spleens from mice 5 days after MCAO in comparison to sham-operated controls. Each probeset is represented in a blue-red row Z-score with blue indicating low expression and red indicating high expression of the probeset (sham $n=2$, MCAO $n=3$). (**b**) Some of the most down-regulated transcripts identified 5 days after MCAO in comparison to sham-operated controls are normally highly expressed in B-cell populations and are essential for normal development and/or function. (**c**) Transcripts differentially expressed in spleens after MCAO were analysed for enrichment ($P<0.05$ with Benjamini correction) of Gene Ontology (GO) Biological Process in DAVID which identified GO terms associated with immune system development and activation. (**d**) Pathway analysis in KEGG of transcripts down regulated after MCAO identified genes involved in multiple stages of the mature B-cell development pathway. (**e**) Microarray expression levels show transcripts for receptors important for the provision of survival signals expressed on B cells (*Tnfsfr13c*, *Tnfsfr13b* and *Cd40*,) are reduced after MCAO. In contrast, expression of the ligands for these (*Tnfsf13b* and *Tnfsf13*) which are expressed by resident stromal cell populations remain unchanged/ increased. Data show mean + s.d.; NS, not significant; *$P<0.05$; **$P<0.01$; ***$P<0.001$; unpaired *t*-test.

significantly lower in patients who developed infection within the first 2 weeks in comparison to those who did not (Fig. 5b). The baseline characteristics of patients with or without infections, and the matched controls, are shown in Supplementary Table 1. In agreement with previous studies showing an association between initial stroke severity and risk of infection, stroke severity (NIHSS) on presentation was significantly higher in patients

who developed infection within the first 2 weeks after stroke (Fig. 5c). We performed an exploratory analysis to further assess the potential relationship among stroke severity, IgM concentration and infection after stroke. This showed a trend indicating that the majority of infections occurred in patients with a combination of high stroke severity and low IgM concentration (Fig. 5d). Thus, our data from both acute stroke

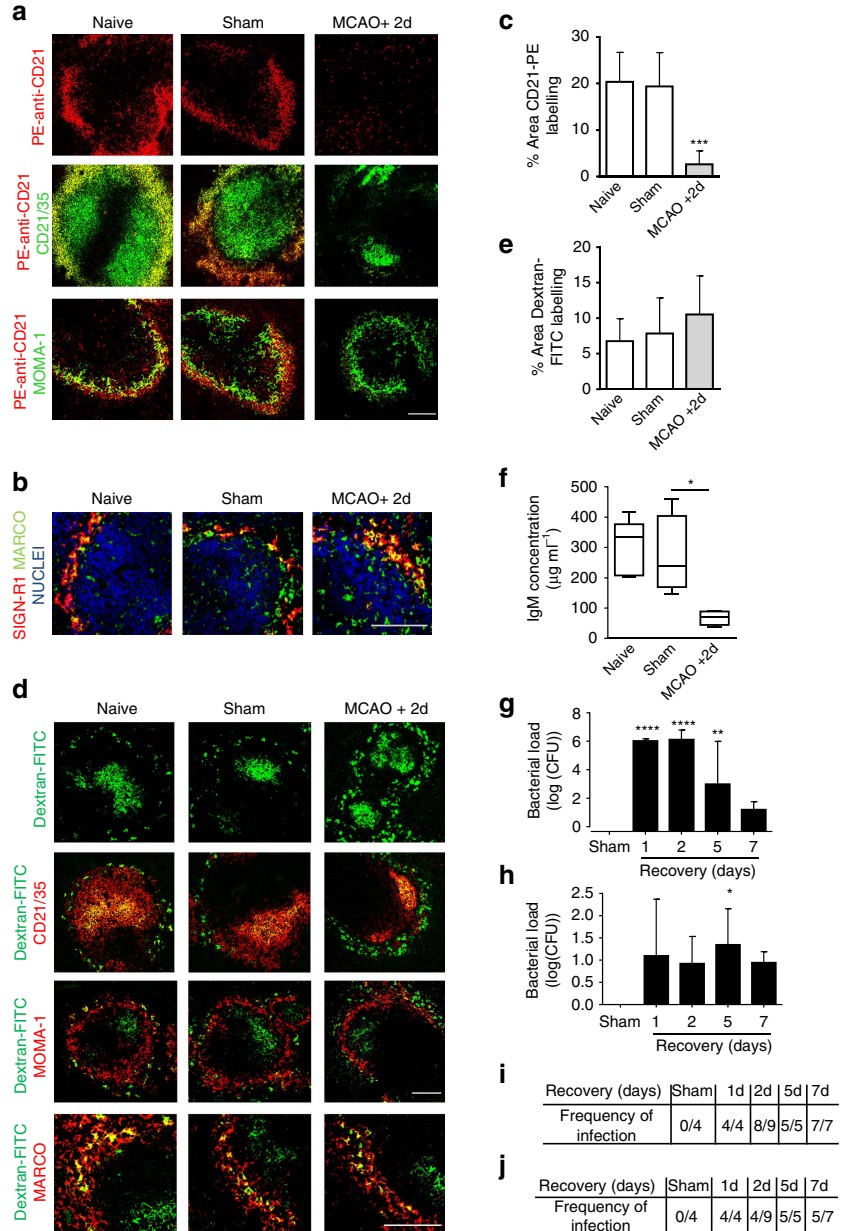

**Figure 4 | Loss of MZ B-cell functions and bacterial infection after experimental stroke.** (**a**) Fluorescently labelled model antigen (red, PE-anti-CD21) is trapped by MZ B cells in the spleens of naïve and sham-operated animals 1 h after i.v. injection however little is detected in the MZ of mice recovered 2 days after MCAO. (**b**) Immunolabelling demonstrates that the loss of MZ B cells (as described Fig. 2) at 2 days after MCAO does not result in a loss of expression of the c-type lectin SIGN-R1 (red) by MZ macrophages (green, MARCO). (**c**) Quantification of the percentage area of PE-anti-CD21 per 846.28 $\mu m^2$ image confirms a significant reduction of PE-anti-CD21 trapping in mice 2 days after MCAO in comparison to sham-operated animals. (**d**) MZ macrophage-targeted antigen (green, Dextran FITC) is trapped by MARCO$^+$ MZ macrophages in the spleens from animals recovered from both MCAO and sham surgery and in naïve controls at 1 h post injection (Supplementary Fig. 4). Dextran-FITC remains associated with MZ macrophages (red, MARCO) 24 h post injection and has efficiently transported to the follicle where it co-localises with FDC networks (red, CD21/35) in all experimental groups. (**e**) Quantification of the percentage area of Dextran-FITC per 846.28 $\mu m^2$ image confirms there is no significant difference in Dextran-FITC trapping in spleens from mice 2 days after MCAO in comparison to sham-operated controls (naïve $n = 3$, sham $n = 3$, MCAO $n = 6$). (**f**) Reduced concentration of circulating IgM 2 days after MCAO in comparison to sham-operated controls (naïve $n = 4$, sham $n = 4$, MCAO $n = 9$). Spontaneous bacterial infection occurs in both the lung (**g**) and blood (**h**) of mice from 1 day after MCAO onwards but cannot be found in sham-operated cage mates (sham $n = 4$, 1d $n = 4$, 2d $n = 9$, 5d $n = 5$, 7d $n = 7$). Description of the frequency of infection in the lung (**i**) and blood (**j**) of mice for each recovery time point. Scale bars (**a,d**; top panel) 200 $\mu m$; (**b,d**; bottom panel) 50 $\mu m$. Data show (**f**) median ± min to max value + s.d., *$P < 0.05$ Mann–Whitney test; (**c,e,g,h**) mean + s.d., One-way ANOVA with Bonferroni correction comparing all groups to sham *$P < 0.05$, **$P < 0.01$, ***$P < 0.001$.

patients and an experimental model of stroke show a common effect on IgM indicative of stroke-induced impairments of systemic innate-like B-cell-dependent pathways predisposing to infection.

**Sensitivity of mature and MZ B cells to β-adrenergic stimulation.** Previous studies have highlighted the role of neural pathways in immunosuppression after stroke. In particular, autonomic circuits involving adrenergic signalling have been implicated in impaired

T-cell function and bacterial infection after experimental stroke and altered systemic catecholamine metabolite levels were associated with infection and mortality in stroke patients[2,38]. The effects of adrenergic signalling on B cells after stroke are unknown, however the rapid effects of MCAO on B-cell composition and function in the present study suggested neural signalling as a plausible underlying mechanism. The concentration of splenic noradrenaline was significantly increased by 1 day after MCAO and remained elevated up to 5 day in comparison to sham-operated controls (Fig. 6a) whereas adrenaline concentrations were unaltered (Fig. 6b). β2-adrenergic receptor (β2-AR) surface protein expression was detected on both MZ and follicular B cells with a significantly higher proportion of MZ B cells expressing the receptor (Fig. 6c). This was despite a small but significantly lower expression of *Adrb2* on MZ B cells (Supplementary Fig. 5), consistent with Immgen expression data (https://www.immgen.org/) although differences between gene

and surface protein abundance are not uncommon due to post-transcriptional and post-translational modification including regulation of protein trafficking and degradation[39]. Stimulation of *in vitro* cultures of isolated splenocytes with noradrenaline resulted in a concentration-dependent reduction in B-cell viability with a greater reduction in viability of MZ B cells (37%) than in follicular B cells (13%; Fig. 6d). These data are consistent with our *in vivo* observations on lymphoid tissue changes after MCAO where both MZ and follicular B cells numbers are decreased but MZ B cells are affected to a greater extent. The more prevalent surface protein expression of β2-AR on MZ B cells may underlie this relatively greater sensitivity.

The β2-AR was also expressed on bone marrow B-cell populations in a subset-specific profile (gating strategy presented in Supplementary Fig. 2c). The proportion of B220[+]CD23[+]CD21[+] mature naïve recirculating and CD138[+]B220[+]CD21[−/lo] plasma B-cells expressing the β2-AR was markedly greater than B220[+]CD23[−]CD21[+] immature and CD138[+]B220[−]CD21[−/lo] memory B cells where negligible expression was apparent (Fig. 6e). Analysis of β2-AR expression on more discrete subsets of bone marrow B-cells using a more extensive antibody cocktail and alternative gating strategy showed the same expression pattern (Supplementary Fig. 6). This suggests β2-AR expression may be developmentally regulated with expression induced upon B-cell maturation and down-regulated upon differentiation to a long-lived, memory phenotype. The relevance to stroke-related B-cell dysfunction was evident in the changes in bone marrow B-cell subpopulations after MCAO. Mature recirculating subpopulations comprising naïve and plasma B cells were significantly reduced, whereas immature B cells newly differentiated from haematopoietic precursors and long-lived memory B cells remained relatively unaffected (Fig. 6f,g). Thus, the sensitivity of bone marrow, as well as splenic B cells, to MCAO-induced loss corresponds with the pattern of β2-AR expression across different subsets. These data also suggest that the sustained loss of splenic B cells up to 5 day after MCAO is unlikely due to deficits in cellular repopulation of lymphoid tissues as the numbers of immature progenitor B cells in the bone marrow were unaffected. In view of the similar effects of stroke on IgM in patients and mice we assessed if human B cells expressed β2-AR. β2-AR expression was detected on 85% of CD19[+] B cells in blood from healthy volunteers (Fig. 6h,i) suggesting that human B cells could be similarly sensitive to increased catecholamine levels after stroke.

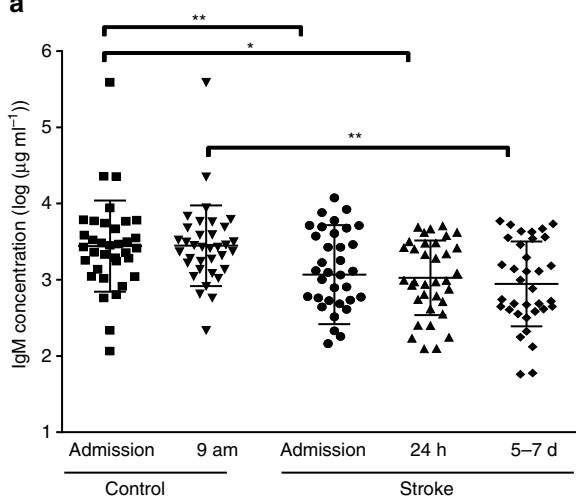

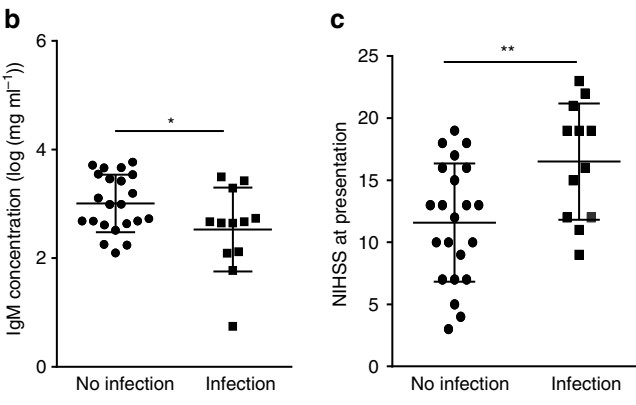

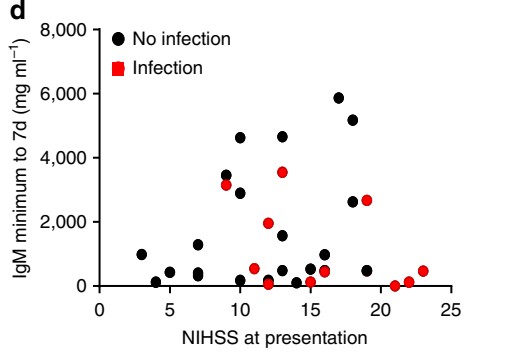

**Figure 5 | Circulating IgM concentration is suppressed early after stroke in patients.** (**a**) Comparison of log-transformed plasma IgM concentrations shows a significantly reduced mean IgM concentration in patients at admission (up to 12 h after onset of stroke symptoms), 24 h and 5–7 days after stroke in comparison to paired controls. To control for circadian variation, stroke samples taken at admission and 24 h were compared to paired admission controls, and stroke samples taken at 5–7 days were compared to 09:00 hours paired controls ($n = 38$). (**b**) Minimum IgM concentration measured in the first 7 day after stroke, was significantly lower in stroke patients who developed infection within 14 days after stroke onset than in those who did not ( + infection $n = 12$, − infection $n = 23$). (**c**) Average stroke severity on presentation, as measured by National Institute of Health Stroke Scale (NIHSS) was significantly higher in patients who developed infection within 14 days post stroke in comparison to those who did not ( + infection $n = 12$, − infection $n = 23$). (**d**) Analysis of the relationship between minimum IgM concentration measured in the first 7 day after stroke and stroke severity (NIHSS score) highlighted a trend for patients who develop infection within 14 day after stroke to have a combination of higher stroke severity and low minimum IgM concentration. Data show mean ± s.d., (**a**) Paired one-way ANOVA with Bonferroni correction; (**b,c**) unpaired *t*-test; *$P < 0.05$, **$P < 0.01$.

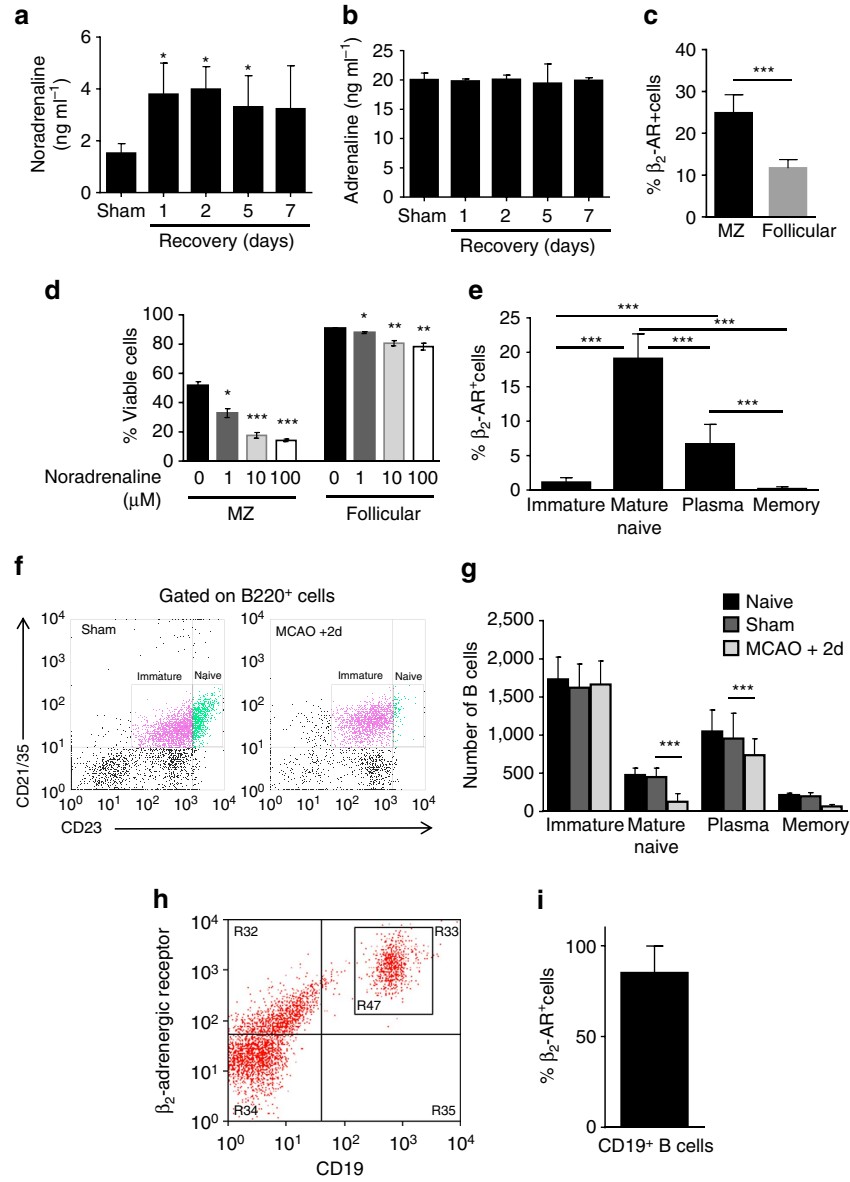

**Figure 6 | Selective sensitivity of mature and MZ B cells to β−adrenergic stimulation.** Homogenate from equal weights of spleen tissue was prepared to measure catecholamine levels. (**a**) Splenic noradrenaline concentration is significantly increased 1–5 days after MCAO in comparison to sham-operated controls. (**b**) No difference in splenic adrenaline concentration was detected (sham $n = 4$, 1d $n = 4$, 2d $n = 9$, 5d $n = 5$, 7d $n = 7$). (**c**) Splenocyte suspensions were prepared from naïve C57BL/6 mice and labelled to differentiate B-cell subsets (Supplementary Fig. 2b) and expression of β2-AR by flow cytometry to determine the percentage of β2-AR$^+$ MZ and follicular B cells ($n = 12$, 4 biological replicates with 3 technical replicates each). (**d**) Mixed splenocytes were cultured with increasing concentrations of noradrenaline for 4 h and viability of B cells was determined by flow cytometry using B-cell markers described previously (Supplementary Fig. 2b) and Alexa Fluor-488-Annexin V dead cell apoptosis kit. Noradrenaline significantly reduced viability of MZ and follicular B cells in a dose-dependent manner ($n = 3$ biological replicates). (**e**) Bone marrow cells were prepared from naïve C57BL/6 mice and labelled to determine B-cell subset expression of β2-AR (Supplementary Fig. 2c). Levels of β2-AR were higher on mature recirculating naïve B220$^+$CD23$^+$CD21$^+$ and CD138$^+$B220$^+$CD21$^{-/lo}$ plasma B cells. Negligible expression of β2-AR was detected on B220$^+$CD23$^-$CD21$^+$ immature and CD138$^+$B220$^-$CD21$^{-/lo}$ long-lived memory B cells ($n = 12$, 4 biological replicates with 3 technical replicates each). (**f**) Bone marrow cells were gated on B220 and plotted against CD23 and CD21/35 to distinguish naïve and immature B cells. Cells were also gated to distinguish plasma from memory B cells (Supplementary Fig. 2c). (**g**) MCAO-induced significant reductions in B220$^+$CD23$^+$CD21$^+$ naïve and CD138$^+$B220$^+$CD21$^{-/lo}$ plasma B cells (sham $n = 4$, MCAO $n = 6$). B220$^+$CD23$^-$CD21$^+$ immature and CD138$^+$B220$^-$CD21$^{-/lo}$ memory B cells were not significantly altered. (**h**) Healthy human donor blood was labelled for flow cytometry analysis of β2-AR expression on CD19$^+$ B cells. (**i**) 80% of CD19$^+$ B cells in human blood express β2-AR ($n = 12$, 4 biological replicates with 3 technical replicates). Data show mean + s.d. (**b,c,e**) One-way ANOVA with Bonferroni correction, (**f**) One-way ANOVA with TUKEY correction (**d,h**) unpaired t-test *$P < 0.05$, **$P < 0.01$, ***$P < 0.001$.

**Blockade of β-adrenergic signalling prevents loss of MZ B cells.** We next determined if blocking β-AR signalling *in vivo* could prevent the loss of MZ B cells induced by MCAO using the pan-β-AR blocker, propranolol. Propranolol treatment resulted in preservation of splenic architecture and retention of both MZ and follicular B cells after MCAO in comparison to PBS-treated mice (Fig. 7a). Quantification of immunolabelling confirmed a significantly greater area of MZ B-cell labelling using multiple MZ

B-cell markers (CD21/35, CD1d and IgM (Fig. 7b). The area of total B220$^+$ B-cell labelling in WP was also significantly greater in animals treated with propranolol in comparison to PBS-treated controls (Fig. 7c). In conjunction with preservation of MZ B cells, plasma IgM concentration after MCAO was significantly greater in propranolol-treated mice compared to PBS-treated controls

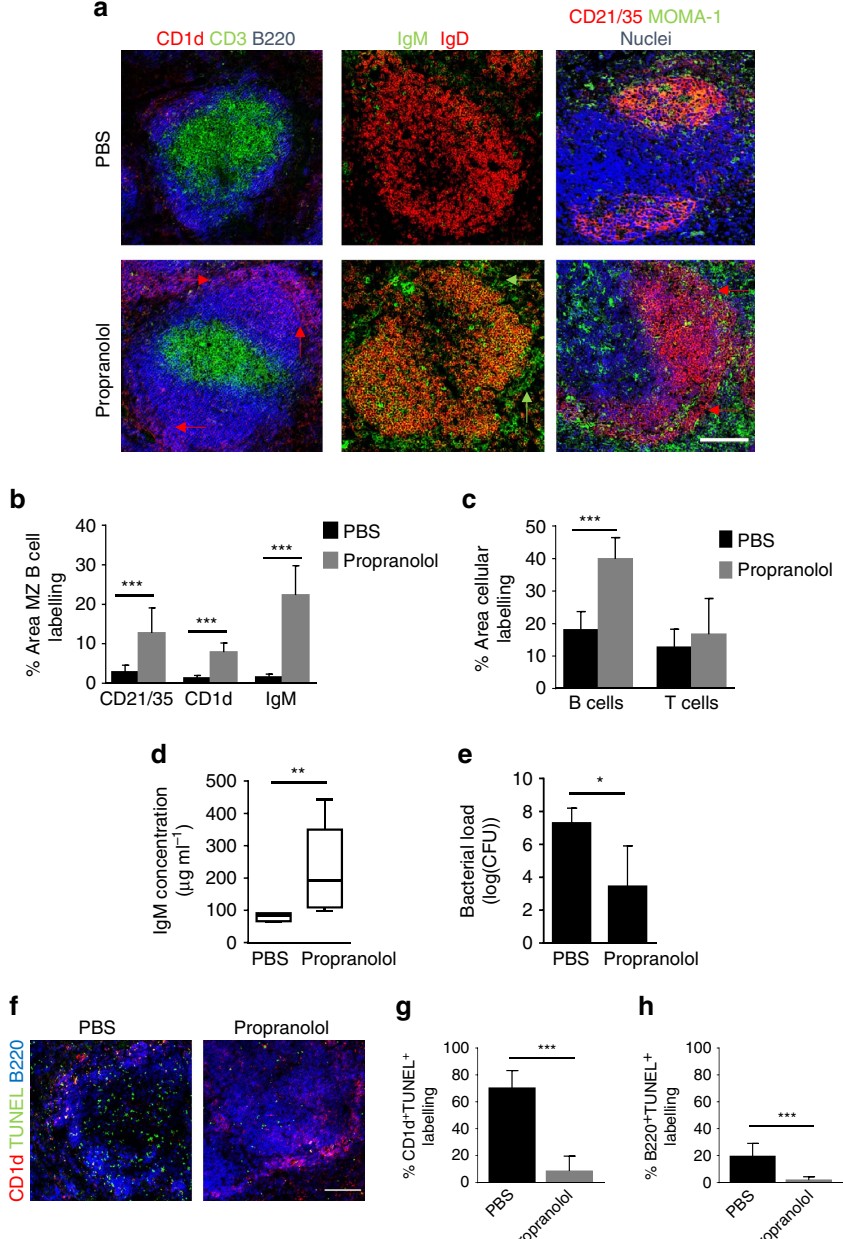

**Figure 7 | Blockade of β−adrenergic signalling prevents loss of MZ B cells.** To block β−adrenergic signalling, animals were treated with 30 mg kg$^{-1}$ propranolol or PBS vehicle control immediately before MCAO and again 4 h later and recovered for 2 days before analysis. (**a**) Immunolabelling of spleens demonstrates propranolol treatment after MCAO preserves splenic architecture with organized B-cell follicles (blue, B220) and T-cell zones (green, CD3).The preservation of MZ B cells after propranolol treatment can also be detected via CD1d (red), IgM (green) and CD21/35 (red) immunolabelling. PBS-treated controls show loss of MZ and follicular B cells and disrupted microarchitecture after MCAO as described previously (Figs 1 and 2; PBS n = 4, Propranolol n = 6). (**b**) Quantification of percentage area of MZ B-cell immunolabelling per 846.28 μm$^2$ image using MZ B-cell markers (CD21/35, CD1d and IgM) confirmed a significant increase in the area of MZ B-cell labelling after propranolol treatment in comparison to PBS-treated controls. (**c**) Similar quantification showed a significant increase in the percentage area of B220$^+$ B-cell labelling in propranolol-treated mice in comparison to PBS-treated controls but no significant effect on the area of T-cell labelling. (**d**) Plasma IgM concentration was significantly greater in propranolol-treated mice in comparison to PBS-treated controls. (**e**) Lung bacterial burden 2 days after MCAO is significantly reduced in propranolol-treated animals in comparison to PBS-treated controls. (**f**) Spleens from animals treated with propranolol after MCAO have decreased death (green, TUNEL) of follicular (blue, B220) and MZ B cells (red, CD1d) in the WP in comparison to those treated with propranolol. Quantitative analysis of the percentage of (**g**) CD1d$^+$ B220$^+$ MZ B cell or the percentage of (**h**) CD1d$^-$ B220$^+$ follicular B-cell immunolabelling that co-localizes with TUNEL as a measure of MZ B or follicular B-cell death, shows a reduction in both CD1d$^+$ B220$^+$ TUNEL$^+$ and CD1d$^-$ B220$^+$ TUNEL$^+$ cells in animals treated with propranolol after MCAO in comparison to those treated with PBS. (PBS n = 4, Propranolol n = 6). Scale bar (**a,f**) 200 μm. Data show (**b,c,e,g,h**) mean + s.d.; unpaired t-test, (**d**) median ± min to max value, Mann–Whitney test. *P < 0.05, **P < 0.01, ***P < 0.001.

(Fig. 7d) and levels approximated those observed in sham-operated mice described above (Fig. 4f). Propranolol treatment significantly reduced bacterial burden in comparison to PBS-treated controls (Fig. 7e). Protection of B-cell numbers in propranolol-treated mice was mediated by prevention of MZ and follicular B-cell death as determined by a reduction in MZ (Fig. 7f,g) and follicular (Fig. 7f,h) B-cell immunolabelling co-colocalising with TUNEL in comparison to PBS-treated controls. There was no significant effect of propranolol on infarct volume although we did note a trend towards smaller infarcts in propranolol-treated mice (Supplementary Fig. 7a). However, the extent of brain injury was comparable to that observed above (Supplementary Fig. 1a) that induced severe splenic disruption which was absent in propranolol-treated mice. Moreover, there was no correlation between infarct volume and plasma IgM concentration (Supplementary Fig. 7a,b) yet lung bacterial burden inversely correlated with plasma IgM concentration. This supports that rescue from severely suppressed levels of IgM in propranolol-treated mice is associated with lower susceptibility to infection (Supplementary Fig. 7c). Together, these data suggest that propranolol can affect plasma IgM concentration independently of the extent of primary brain injury and most likely through preservation of splenic B-cell populations after MCAO, notably MZ B cells, the major source of rapidly inducible IgM. Collectively with *in vitro* findings above, these data show that β-AR signalling mediates loss of MZ B cells and their innate-like functions early after MCAO. Importantly, reversal of these MCAO-induced impairments by β-AR blockade is associated with reduced susceptibility to stroke-associated infection.

## Discussion

The data presented here provide compelling evidence of neuroimmune driven disruption to innate-like B cells after stroke and suggest these alterations as a novel mechanism contributing to the infection-prone state induced by ischaemic stroke in addition to previously described impairments in other immune cell populations. To our knowledge, this is the first demonstration that anti-microbial function of innate-like B cells is affected by stroke. Key findings include that (1) stroke induces marked loss of splenic MZ B cells along with gross disruption to splenic microarchitecture, (2) MZ B-cell capture of antigen is deficient after stroke, (3) circulating IgM levels are suppressed after stroke in mice and humans and lower IgM levels are evident in stroke patients with infection and (4) β2-AR-mediated signalling drives stroke-induced MZ B-cell loss and susceptibility to infection.

MZ B cells are one of several types of lymphocytes that display innate-like attributes as reflected by their innate activation via pattern recognition receptors, speed of responsiveness and antigen polyreactivity, making them vital immune sentinels that mediate rapid systemic anti-microbial immunity[17,19]. The depletion of MZ B cells as early as 24 h after MCAO results in deficits in the ability to capture MZ B-cell-targeted antigen and a high bacterial burden accumulates in the lungs over a similar time-frame. This rapid loss of a cell population which is critical for early anti-bacterial defence is highly relevant because of the timing of infections after stroke. Indeed, almost all cases of pneumonia occur within the first week, and half of these within 2 days[1,4]. Such an immediate window of susceptibility suggests that deficiencies in innate or innate-like functions are important. Previous studies have reported altered macrophage responses early after stroke, however these have largely focussed on markers of immunoregulatory phenotype or antigen presentation rather than direct effector and microbicidal functions[5,40–42]. In addition to their speed of response, MZ B cells are specialised to respond to T-independent antigens that include polysaccharides of encapsulated bacteria[17]. The bacterial strains causing early onset pneumonia in stroke patients most frequently include the encapsulated *Streptococcus pneumoniae*, *Haemophilus influenzae*, *Klebsiella pneumoniae* and *Staphyloccocus aureus*[43–45]. Loss of splenic MZ B cells early after MCAO is therefore of particular relevance to both the types of bacterial infection and when they occur in stroke patients.

Splenic MZ B cells are a major source of IgM, particularly during the early stages of infection[16,46–49]. Despite a high bacterial burden in the lungs and blood after MCAO and hence strong stimulus for antibody secretion, we found significantly lower levels of circulating IgM. IgM is also produced constitutively as 'natural' antibody by MZ B cells and B1 B cells[50]. Lower IgM levels after MCAO may reflect a loss of constitutive natural antibody production as well as impaired infection-triggered production due to a lack of MZ B cells. IgM has a short half-life ($\sim$12 h) thus the early reduction in IgM we observed after MCAO is consistent with this rate of turnover[51]. In agreement with the mouse data, which showed rapid and sustained splenic MZ disruption, plasma IgM concentration was also significantly reduced in patients as early as 24 h after stroke and this was maintained up to the 5–7 days time point. Although it is possible that individuals who subsequently have stroke could have lower pre-existing IgM levels than those who do not, this is an unlikely explanation given that paired controls were well matched for age, sex and degree of atherosclerosis. A significantly greater reduction in IgM concentration during the first week after stroke was associated with infection, which was observed in one third of patients. As described in previous studies, infection was also associated with greater initial stroke severity[4,12,52]. Our data are consistent with a recent study showing lower IgG levels in stroke patients which was associated with bacterial infection, however, these deficits were not associated with a B-cell-dependent mechanism[53]. In conjunction with our mouse MCAO data, it is reasonable to speculate that deficits in innate-like B-cell function, including IgM production, could mediate the effects of greater stroke severity on infection risk in patients. However the precise mechanisms of reduced circulating IgM concentrations leading to susceptibility to bacterial lung infection still remain to be fully elucidated in the context of stroke. Our exploratory analysis indicated a combination of greater stroke severity and lower IgM may identify those individuals at particularly high risk of infection although further studies are warranted to address these links more definitively.

The loss of MZ B cells and suppression of circulating IgM levels after MCAO were reversed by blockade of β-AR signalling and this was accompanied by a significantly lower bacterial lung burden. The spleen is directly innervated and our data showing elevated splenic noradrenaline levels after MCAO are consistent with previous studies reporting activation of sympathetic neural circuits after MCAO and increased circulating catecholamines (or metabolites of these) in stroke patients[2,54]. The expression of the β2-AR across different B-cell subsets was previously unknown and our data provide a rational explanation for the relatively greater sensitivity of MZ B cells to depletion after MCAO. Expression of the β2-AR on human B cells further supports their sensitivity to similar neural signals as observed in mice. Of particular note is that few immature B cells expressed the β2-AR and this correlated with stable numbers in the bone marrow after MCAO. Loss of specific splenic B-cell populationsis therefore unlikely to be the result of impaired B-cell differentiation in bone marrow and moreover highlights selective sensitivity of B cells at different stages of development to neural signalling activated after stroke. However we do not exclude that noradrenaline could also act indirectly via an intermediate cell to affect B cells after experimental stroke and the precise mechanisms of how brain

ischaemia results in increased splenic noradrenaline remain to be determined. We used propranolol simply as an experimental tool to antagonise $\beta_2$-AR signalling and it is likely that therapeutic targeting of B cells themselves would provide a superior approach clinically to target infection susceptibility given the potential adverse effects of general $\beta_2$-AR blockade. Nonetheless it is relevant to note recent interest in re-evaluating the role of $\beta$-blocker treatment in stroke patients[55].

As noted above, our data implicate B-cell death as the terminal mechanism responsible for their depletion after MCAO. $\beta_2$-AR activation on B cells has thus far only been studied in vitro and reported effects vary dependent on other exogenous stimuli included in culture[56]. Atrophy of lymphoid organs including the spleen occurs after experimental stroke and is thought to reflect general apoptosis of lymphocytes although effects on MZ B cells specifically were unknown[40]. Previous studies have suggested that surgical and/or anaesthetic stress could contribute to transient immune alterations after MCAO[57,58]. However effects on MZ B cells after MCAO were maintained out to 7 day and we observed no alterations in cell death or splenic microarchitecture and did not detect any bacterial growth in sham-operated mice. Thus, the effects we describe are specific to the stroke-induced brain injury.

The disruption to systemic anti-microbial B-cell function after stroke we describe here raises the possibility that interventions directed at preventing these changes could be a novel strategy to reduce infection risk and associated mortality/morbidity after stroke. Currently, there are no treatments proven to reduce rates of infection after stroke that improve clinical outcome, including preventative antibiotic therapy. A recent study suggests this may in part be due to ineffective prevention of pneumonia[59–61]. Transiently augmenting systemic B-cell function, in particular innate-like responses in the first few days after stroke, may offer potential advantages to other approaches. Effects of B cells in the brain during the acute phase of experimental stroke are neuroprotective and limit tissue-damaging inflammation suggesting the risk of exacerbating brain damage may be relatively low for B-cell-targeted approaches[62,63]. The recent discovery that auto-antibody production by B cells may promote dementia long-term after experimental stroke, and have pathological effects in models of CNS trauma must be considered, however, these effects are likely greatly separated in time and space from the role of anti-microbial innate-like B-cell function relevant to infection control during the acute phase of stroke[64,65]. Although increased risk of autoimmunity must also be considered, a recent study showed that generally attenuating stroke-induced immunosuppression increased levels of CNS-specific autoantibodies but had no detrimental effect on long-term outcome in mice[66].

In conclusion, we propose that studies examining the effects of B-cell-targeted approaches to preventing infection after stroke are warranted and suggest that markers of systemic B-cell function, as an adjunct to existing indicators, may have potential utility in identifying stroke patients at particularly high risk of infection.

## Methods

**Animals.** Male 8–10-week-old C57BL/6 J mice were purchased (Charles River Laboratories, UK) and used in all experiments unless otherwise stated (Fig. 1h,i and Supplementary Fig. 1 batch numbers 4,551, 4,848, 4,947; Figs 1a–g,2a–c,e–g, 3a–e,4f–j and 6a,b, Supplementary Fig. 3 batch number 5,461; Fig. 4a–e and Supplementary Fig. 4 batch numbers 9,224, 9,225, 9,777, 9,778; Fig. 7 and Supplementary Fig. 7 batch number 6,593 and Fig. 6c–e and Supplementary Figs 5 and 6, batch number unavailable). $Ccr2^{RFP/+}$ transgenic mice were used for flow cytometric analysis of B-cell subsets after stroke (Figs 2d and 6f,g and Supplementary Fig. 2) and have been previously described[67]. Breeding pairs of $Ccr2^{RFP/+}$ reporter mice on a C57BL/6 J background were purchased from Jackson Laboratories (ME, USA) and a colony was established in house. Mice were maintained under specific pathogen free (SPF) conditions and a standard 12 h light/dark cycle with unrestricted access to food and water. Mice were housed in

individually ventilated cages in groups of up to 5 mice and were acclimatized for a minimum of 1 week before procedures. All animal experiments were carried out under the authority of a UK Home Office Project Licence in accordance with the 'Animals (Scientific procedures) Act 1986' and Directive 2010/63/EU and were approved by both The Roslin Institute's and the University of Edinburgh's Animal Welfare and Ethics Review Board. Experimental design, analysis and reporting followed the ARRIVE guidelines (https://www.nc3rs.org.uk/arrive-guidelines).

**Experimental stroke model.** MCAO was performed under isoflourane anaesthesia (with $O_2$ and $N_2O$) by insertion of a 6-0 nylon monofilament with a 2-mm coated tip (210 μm diameter; Doccol, USA) through the external carotid artery and advanced through the internal carotid artery to occlude the MCA. The filament was withdrawn after 40 min to allow reperfusion, the neck wound sutured and the animals recovered. Topical local anaesthetic (lidocaine/prilocaine) was applied to the wound. For sham surgery, the filament was advanced to the MCA and immediately retracted. Sham-operated animals remained anaesthetized for 40 min and recovered as above. Core body temperature was maintained at $37 \pm 0.5$ °C throughout the procedure with a feedback controlled heating blanket (Harvard Aparatus). Animals were recovered for 1–7 days then anaesthetized and anti-coagulated (3.8% w/v tri-sodium citrate) cardiac blood samples were taken followed by transcardiac perfusion with saline.

**Assessment of infarct volume.** Coronal cryosections (20 μm in thickness) were cut at 400 μm intervals and stained with cresyl violet. Images were captured and infarct area on each coronal section was measured using ImageJ (http://rsb.info.nih.gov/ij/). Infarct volume was calculated by the sum of all areas multiplied by the distance between each section. The volume of damage was corrected for oedema, as calculated by subtracting the volume of the contralateral hemisphere from the ipsilateral hemisphere and expressing the difference as a percentage of the contralateral hemisphere, as previously described[68].

**Immunohistochemistry and analysis of spleens.** Serial frozen sections of spleen (6 μm in thickness) were cut on a cryostat, fixed in ice-cold acetone for 10 min, washed in 0.05% BSA in PBS and blocked using species-specific normal serum (Jackson Immunoresearch Laboratories Inc., PA, USA) according to secondary antibody used. Primary antibodies were incubated for 1 h at room temperature. Follicular B cells were detected using monoclonal antibody (mAb) B220 (2.5 μg ml$^{-1}$) to detect CD45R (Caltag, Towcester, UK), or mAB 11-26c.2a (5 μg ml$^{-1}$) to detect IgD (Biolegend, London, UK). MZ B cells were detected using mAb 1B1 (10 μg ml$^{-1}$) to detect CD1d (BD Biosciences, PharMingen, Oxford, UK), or mAb Il/41 (2.5 μg ml$^{-1}$) to detect IgM (affymetrix eBioscience, Hatfield, UK). MZ B cells and FDCs were visualised with mAb 7G6 (2.5 μg ml$^{-1}$) to detect CR2/CR1 (CD21/CD35; BD Biosciences PharMingen). T cells were detected using mAb 145-2C11 (5 μg ml$^{-1}$) conjugated to biotin to detect CD3 (affymetrix eBioscience). Splenic macrophage populations were visualised by staining with F4/80 (1 μg ml$^{-1}$; clone BM8, Biolegend), MOMA-1 (5 μg ml$^{-1}$) to detect CD169 (Siglec 1; clone 3D6.112, Biolegend), MARCO (5 μg ml$^{-1}$; clone ED31, AbD Serotec, Kidlington, UK), or SIGN R1 (5 μg ml$^{-1}$; CD209, clone 22D1; eBioscience). Splenic proteins BAFFR (Tnfsfr13c; 5 μg ml$^{-1}$) and CD40 (5 μg ml$^{-1}$) were detected using clone 7H22-E16 (Biolegend) and clone 3/23 (Thermo Fisher) respectively. Following the addition of primary antibody, sections were washed in PBS–BSA and 2.5 μg ml$^{-1}$ of species-specific secondary antibody coupled to Alexa Fluor 488, Alexa Fluor 594 dyes or Alexa Fluor 647 fluorochromes (Life Technologies Ltd, Paisley, UK) were incubated for 1 h at room temperature. Sections were washed in PBS–BSA and mounted in fluorescent mounting medium (Dako, Cambridgeshire, UK) and images captured using a Zeiss LSM5 confocal microscope (Zeiss, Welwyn Garden City, UK).

Digital images were analysed using ImageJ software. All spleens from each experimental group were analysed. From each spleen, two sections, 100 μm apart were studied and on each section data from 4 individual areas of WP collected. Fluorescent intensity thresholds were applied and the number of pixels of each colour (black, red, green and yellow) were automatically counted as previously described[69] and used to determine the area of immunolabelling for each cell type. To analyse MZ specific immunolabelling, a border was firstly drawn to separate the MZ from the follicle using MZ macrophage or follicular B-cell immunolabelling as a guide. The area of the WP was excluded and area of MZ zone specific immunolabelling was determined as previous.

**TUNEL labelling of cell death.** Apoptotic TUNEL$^+$ cells were measured using the ApopTag Fluorescein Direct In Situ Apoptosis Detection Kit (Merck Millipore, Hertfordshire, UK) to detect DNA fragmentation according to the manufacturers protocol along with B220$^+$ immunolabelling to distinguish the WP. In brief, immunolabelling for B220 was performed as described above. Sections were then fixed in 1% paraformaldehyde in PBS, washed in PBS and post-fixed in ice-cold ethanol: acetic acid (2:1). Sections were incubated in reaction mix containing DIG-labelled dUTP and terminal deoxynucleotidyl transferase for 60 min. After washing, fluorescein-conjugated anti-DIG was applied to sample, which was then incubated in the dark for 30 min. Sections were washed and mounted in fluorescent

mounting media (Dako) and images captured using a Zeiss LSM5 confocal microscope (Zeiss).

**Flow cytometry.** Mouse spleens were mechanically dissociated and bone marrow obtained by flushing the femur and tibia from the left side in 5 ml PBS containing 0.1% BSA. Anti-coagulated venous blood was collected by venipuncture from healthy human donors. Red blood cells were lysed and resulting single-cell suspensions were used for flow cytometric analysis of B-cell populations and $\beta_2$-AR expression. Briefly, cell surface staining to allow the detection of B-cell subpopulations was carried out using antibodies against CD45R (clone RA3-6B2; Biolegend; 2.5 µg ml$^{-1}$), CD21/35 (CR1/2, clone 4E3; affymetrix eBioscience 1 µg ml$^{-1}$), CD23 (FceRII, clone B3B4; Bioloegend, 1.25 µg ml$^{-1}$), CD138 (syndecan-1, clone DL-101; Biolegend 5 µg ml$^{-1}$), CD43 (clone S11; Biolegend 1 µg ml$^{-1}$), CD93 (clone AA4.1; Biolegend 1 µg ml$^{-1}$), IgD (clone 11-26c.2a; Biolegend 2.5 µg ml$^{-1}$), IgM (clone RMM1; Biolegend, 1 µg ml$^{-1}$) directly conjugated to fluorochromes and purified antibody against $\beta_2$-AR (clone R11E1; Santa Cruz Biotechnology Inc, Heidelberg, Germany, 1 µg ml$^{-1}$). Mouse cells were incubated in anti-mouse CD16/32 (clone 93; Affymetrix eBioscience, 0.4 µg ml$^{-1}$) and human cells in Human TruStain FcX (Fc receptor blocking solution, Biolegend, 1/500) to block Fc receptors then primary antibody cocktails for 30 min at room temperature. For $\beta_2$-AR labelling, anti-$\beta_2$-AR antibody was added for 30 min, cells washed in PBS then incubated with goat-anti-mouse antibody conjugated to FITC (Poly 4053, Biolegend, 1 µg ml$^{-1}$) for 30 min. Cells were washed again and then incubated in primary antibody cocktails. Flow cytometry was performed on a Becton Dickinson LSR Fortessa and analysed by Summit software (Dako). Lymphocytes were gated based on FSC and SSC and 20,000 cells in this gate collected for each sample. Quantitative calibration with Countbright absolute counting beads (Life Technologies Ltd) was used according to the manufacturer's protocol to obtain absolute numbers of each cell population.

**Cell sorting and subset-specific qPCR analysis.** Mouse spleens were mechanically dissociated, red blood cells were lysed and resulting single-cell suspensions were labelled to allow distinction of MZ and follicular B-cell subsets using antibodies against CD45R (clone RA3-6B2; BD Biosciences) CD93 (clone AA4.1; Biolegend), CD23 (clone B3B4; Biolegend) and CD21/35 (clone ebio8D9; eBioscience) directly conjugated to fluorochromes. MZ and follicular B cells were sorted on a FACS Aria Illu and immediately processed for RNA extraction using the RNeasy Plus Micro kit (Qiagen) according to manufacturer's instruction. RNA quantity was determined by Nanodrop 1000 (Thermo Fisher Scientific, Renfrew, UK) and reverse-transcribed using Superscript III Reverse Transcriptase according to manufacturer's instructions (Life Technologies). qPCR was performed in a Stratagene Mx3005P (Agilent Technologies) using Platinum SYBR Green qPCR SuperMix-UDG (Invitrogen) and primer pairs *Adrb2*-f 5′-TTGCCAAGTTCGAG CGACTA-3′ and *Adrb2*-r 5′-GATCCACTGCAATCACGGCAC-3′, *Cd1d*-f 5′-CCA GAGCCTTTGTGTACCAGT-3′ and *Cd1d*-r 5′-TTTTGCTGGGCTTCAGATT GTC-3′, and *Actb* primer kit from geNorm Reference Gene Selection Kit (Primerdesign, Eastleigh, UK). qPCR cycles were performed as follows: hot-start denaturation cycle 95 °C for 1 min, 40 cycles of amplification with 95 °C for 15 s, 60 °C for 30 s and 72 °C for 1 min. $\Delta$Ct (Ct of housekeeping gene, *Actb*, minus Ct of gene of interest) was calculated for each B-cell subset. Data are expressed as a reciprocal of this value.

**Microarray.** RNA was isolated from $\frac{1}{4}$ spleens from sham-operated animals or animals recovered from MCAO for 5 days using the Qiagen RNeasy mini kit (Qiagen, Manchester, UK). RNA integrity and quality were verified using Agilent BioAnalyzer 2100 (Agilent Technologies, Santa Clara, CA, USA). RNA was reverse-transcribed to cDNA, amplified and labelled using the Affymetrix terminal labelling kit (Affymetrix,Santa Clara, CA, USA), according to the manufacturer's instructions. Samples were hybridized to MoGene 1.0 ST chips (Affymetrix) using GeneTitan Hybridisation, Wash and Stain kit for WT array plates (Affymetrix), stained, and scanned by Edinburgh Genomics (Edinburgh, UK). Raw data (.cel) files were normalized directly during import for analysis in Partek Genomics Suite (Partek Inc, MO, USA.). Probesets were annotated using MoGene-1_0-st-v1 Probeset Annotations, Release 35. To determine which transcripts were changed due to experimental stroke, normalized datasets were compared in Partek by ANOVA. Enrichment analysis for Gene Ontology terms was performed in DAVID (http://david.abcc.ncifcrf.gov/) by uploading lists of differentially expressed transcripts and selecting the GOTERM_BP_FAT annotation category. Enrichment was assessed using default settings. Pathway analysis was also performed in DAVID using the KEGG tool. Microarray data are deposited in the NCBI GeoDatasets database with the accession number GSE70841.

**Quantitative PCR.** Total RNA of 500 ng prepared for microarray (as above) was reverse-transcribed using superscript III Reverse transcriptase (Life technologies) according to manufacturer's instructions. qPCR was carried out on a Stratagene Mx3005P instrument (Agilent Technologies) using Qiagen RT² SYBR Green Mastermix (Qiagen) and RT² qPCR primer kits for *Cr2*, *Cd40*, *Tnfsfr13c* and *Gappdh* (Qiagen) according to manufacturer's instructions. qPCR cycles were performed as follows: hot-start denaturation cycle 95 °C for 10 min, 40 cycles of

amplification of 95 °C for 15 s and 60 °C for 1 min $\Delta$Ct (Ct of housekeeping gene, *Gapdh*, minus Ct of gene of interest) was calculated for each sample. Data are expressed as the normalised expression ratio ($2^{(-\Delta\Delta Ct)}$) in comparison to sham samples.

**Bacteriological analysis.** Lungs were rinsed in sterile PBS and transferred to a gentleMACS C Tube containing 1 ml sterile PBS and homogenized on a gentle-MACS Dissociator (Miltenyi Biotech, Surrey, UK). Lung homogenates and blood, collected and prepared as described above, were serially diluted, plated onto Columbia agar and horse blood (EOLabs, Bonnybridge, Scotland), incubated at 37 °C for 48 h and bacterial colonies counted.

**Splenic marginal zone antigen capture.** To assess the capture and transport of antigen by MZ cells *in vivo*, naïve mice or mice recovered 48 h after sham or MCAO, were administered either 20 µg of 3,000 kDa Dextran conjugated to FITC (Sigma, Dorset, UK) or 1 µg antibody against CD21/35 conjugated to PE (eBioscience) by intravenous injection. Spleens were removed either 1 h or 24 h after injection and the presence of MZ cell-associated or FDC-associated fluorescent antigen was identified by immunohistochemistry as described above.

**ELISA measurement of mouse plasma IgM and catecholamines.** Plasma samples prepared from mice recovered for 48 h after sham-operation or MCAO were assayed using mouse Ready-Set-Go ELISA kits to detect IgM (affymetrix eBioscience) and 2-Cat (A-N) research ELISA kits to measure noradrenaline and adrenaline (LDN laboratory diagnostics, Nordhorn, Germany). Blood samples obtained by cardiac puncture were incubated at room temperature for 30 min and centrifuged at 1,000 g and supernatant plasma was collected. For measurement of IgM, plasma was diluted 1/10,000 and ELISA carried out according to manufacturer's instructions. For detection of noradrenaline and adrenaline, 50 µl of spleen homogenate prepared from equal weights of spleen tissue was acylated, enzymatically converted and catecholamines measured by ELISA according to manufacturer's instructions.

**Recruitment of stroke patients and plasma IgM measurement.** Blood from healthy volunteers was collected for the analysis of $\beta$-adrenergic receptor expression under ethical approval obtained from the Lothian Research Ethics Committee (11/AL/0168). The current analysis of stored samples from the stroke patient and control cohort previously described[9,36] was approved by the Health Research Authority National Research and Ethics Service Committee (14/EM/ 1117). Clinical evaluation of patients was as described previously and baseline characteristics of patients and controls are described in Supplementary Table 1 (ref. 36). Briefly, patients over 18 years presenting at Salford Royal Hospital, Salford, UK within 12 h of symptom onset of acute ischaemic stroke were eligible. Patients were excluded if there was any improvement in symptoms since onset, the time of onset of symptoms could not be reliably determined, or there was evidence of active malignancy. Control subjects with no history of stroke or transient ischaemic attack, without clinically evident infection necessitating medical treatment, and without a history of cognitive impairment sufficient to interfere with daily life were matched for age ( ± 5 years), sex and degree of atherosclerosis. Written informed consent (or assent from a relative) was obtained for all patients and control subjects. The National Institutes of Health Stroke Scale (NIHSS) score[70] was used to measure stroke severity at presentation. Infections occurring in the 14 day after stroke onset were recorded prospectively during the study period using all available clinical information and original investigation results. Patients with infections preceding stroke onset were excluded from analysis. Venous blood samples were collected from patients at admission (up to 12 h after the onset of stroke symptoms), the next 09:00 hours time point where admission was before 07:00 or after 11:00 hours, 24 hours after admission, and 5–7 days at 09:00 hours. Blood was also drawn from resting control subjects at 09:00 hours and at a time matched to the patient's time of admission if this was before 07:00 or after 11:00 hours. To control for circadian variations stroke patient samples taken at admission and at 24 h were compared to admission-matched, paired controls, whereas stroke patient samples taken at 5–7 days were compared to 09:00 hours paired controls. Blood was collected into heparin-coated tubes and at 1 h after collection centrifuged at 2,000 g for 30 min, at 4 °C. Plasma was separated, frozen and stored at − 70 °C until analysis. IgM concentration was measured using human Ready-Set-Go ELISA kits to detect IgM (Affymetrix eBioscience) according to manufacturer's instructions.

***In vitro* splenocyte stimulation with noradrenaline.** Single-cell suspensions were prepared from spleens from naïve C57Bl/6 J mice as above. Cells were cultured in 96-well plates ($5 \times 10^6$ cells per ml) in media optimized for B-cell survival (RPMI, 1% L-Glutamate, 10% heat-inactivated foetal calf serum, 1% penicillin-streptomycin, 1% sodium pyruvate, 1% non-essential amino acids and 0.1% $\beta$-mercaptoethanol) and incubated with 1, 10 or 100 nM of noradrenaline for 4 h (Sigma Aldrich). Cells were collected for flow cytometry labelling as above using CD45R$^+$ to identify B cells. Cell viability/ death was measured using an Alexa Fluor 488-Annexin-V dead cell apoptosis kit (Life Technology Ltd.) based on

labelling with propidium iodide (PI) and annexin V. Viable cells were negative for both propidium iodide and annexin V labelling. At least three independent experiments each comprising triplicate samples were performed for all *in vitro* experiments.

**In vivo blockade of β-adrenergic receptors.** Treatment regimen was chosen according to the literature[2] to block β-adrenergic signalling pathways and not to perform a pre-clinical study on the effect of β-blockers in stroke. (S)-(−)-Propranolol hydrochloride (propranolol; Sigma Aldrich) (30 mg kg$^{-1}$) was administered by i.p injection immediately before and 4 h after MCAO. An equivalent volume of PBS was administered in vehicle-treated animals. Animals were recovered for 2 d after MCAO and tissues collected as described above.

**Experimental design and statistical analysis.** Sample sizes were estimated from pilot studies and previous data using power analysis or the resource equation method. Animals were randomised to experimental groups (for example, time point after MCAO, drug treatment) using a computer-based random number generator (https://www.randomizer.org/) and drug treatments were administered in a blinded manner. The assessor was unaware of allocation to experimental group during analysis of outcome measures although no formal blinding procedures were in place. Data are presented as mean ± s.d. unless otherwise indicated. For normally distributed data differences were tested using unpaired Student's $t$-test or analysis of variance (ANOVA) with Bonferroni or Tukey correction unless otherwise stated. For non-normally distributed data, equivalent non-parametric tests were used and results displayed as median ± minimum and maximum values. Data analysis for bacterial colonies after MCAO and IgM concentration in patient samples was on log$_{10}$-transformed values. For comparison of IgM concentration in stroke patients with and without infection, minimum IgM concentration was defined as the lowest value up to 7 day after stroke. Analysis of the relationship among stroke severity (NIHSS score), IgM concentration and infection in stroke patients is observational and for exploratory purposes only. Data were analysed using GraphPad Prism. In all experiments values of $P \leq 0.05$ were accepted as statistically significant.

**Data availability.** Microarray data that support findings of this study are deposited in the NCBI GeoDatasets database with the accession number GSE70841. All other data supporting the findings of this study are available from the authors.

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

## Acknowledgements

We thank Dr Sara Clohisey and Dr Kenneth Baillie for provision of healthy human blood samples, Dr Hedley Emsley and Dr Carole Gavin for patient recruitment and data collection during the original stroke patient study, and all blood donors and patients for their participation and consent. We thank staff in the Biological Research Facility (Roslin Institute) for animal husbandry and technical support, Sharon Hulme and Margaret Hoadley for assistance with ethical applications and sample transfer, and Prof Stuart Allan, Prof Steve Hopkins, Prof Nancy Rothwell and Prof Pippa Tyrrell for discussion of data and contributions to the original stroke patient study. This work was funded by grants from BBSRC (BB/J004332/1) and MRC (MR/L003384/1). The Roslin Institute and Edinburgh Genomics are partly supported through core grants from NERC (R8/H10/56), MRC (MR/K001744/1) and BBSRC (BB/J004243/1, BB/J004332/1).

## Author contributions

L.M. and B.W.M. conceived the study, performed experiments, performed analysis and wrote the manuscript. C.J.S. provided access to clinical samples and clinical data and contributed to the analysis of clinical data. All authors discussed the results and edited the manuscript.

## Additional information

**Competing interests:** The authors declare no competing financial interests.

**DOI: 10.1038/ncomms16151**    **OPEN**

# Corrigendum: Adrenergic-mediated loss of splenic marginal zone B cells contributes to infection susceptibility after stroke

Laura McCulloch, Craig J. Smith & Barry W. McColl

Nature Communications 8:15051 doi: 10.1038/ncomms15051 (2017); Published 19 Apr 2017; Updated 18 Aug 2017

The affiliation details for Barry W. McColl are incorrect in this Article. The correct affiliation details for this author are given below:

The Roslin Institute and R(D)SVS, University of Edinburgh, Easter Bush, Midlothian EH25 9RG, UK.

UK Dementia Research Institute, University of Edinburgh, Edinburgh Medical School, 47 Little France Crescent, Edinburgh EH16 4TJ, UK.

