## [Peer Review File · Nature Communications]

REVIEWERS' COMMENTS:

Reviewer #2 (Remarks to the Author):

McCulloch et al demonstrate that experimental ischemic stroke, in a mouse model, leads to reduction of splenic MZ B cells, an effect associated with alteration of the splenic microarchitecture. Stroke reduces the ability of MZ B cell to capture of antigen and reduces circulating IgM levels both in mice and humans. Interestingly, reduced IgM levels are observed in stroke patients with infection. The splenic effects are attributed to β 2-adrenergic receptor-mediated signaling.

The study is novel in that it implicates the depletion of splenic MZ B cells and reduced IGM with post-stroke infections, and suggests a signaling pathway underlying these effects. The findings expand our understanding of the mechanisms of increased susceptibility to infections after stroke. Some questions that remain to be addressed include (a) establishing causality between the splenic findings and reduction in circulating IGM, and the increased infections and (b) elucidating the signaling steps through which brain ischemia initiates the adrenergic response that ultimately targets the splenic cells.

Reviewer #3 (Remarks to the Author):

The author's responses to reviewer 3 are acceptable.

REVIEWERS' COMMENTS:

Reviewer #2 (Remarks to the Author):

McCulloch et al demonstrate that experimental ischemic stroke, in a mouse model, leads to reduction of splenic MZ B cells, an effect associated with alteration of the splenic microarchitecture. Stroke reduces the ability of MZ B cell to capture of antigen and reduces circulating IgM levels both in mice and humans. Interestingly, reduced IgM levels are observed in stroke patients with infection. The splenic effects are attributed to β 2-adrenergic receptor-mediated signaling.

The study is novel in that it implicates the depletion of splenic MZ B cells and reduced IGM with post-stroke infections, and suggests a signaling pathway underlying these effects. The findings expand our understanding of the mechanisms of increased susceptibility to infections after stroke. Some questions that remain to be addressed include (a) establishing causality between the splenic findings and reduction in circulating IGM, and the increased infections and (b) elucidating the signaling steps through which brain ischemia initiates the adrenergic response that ultimately targets the splenic cells.

Response: We thank the reviewer for their further constructive comments and are pleased to note their enthusiasm for the novel findings of our study. We fully agree there are important next steps to take for which our current study forms a basis. It will be important to delineate the direct mechanisms linking splenic MZ disruption, IgM alterations and lung infection, including effects of stroke on the downstream anti-bacterial effector mechanisms – we have included a sentence in the Discussion (p16) to indicate this *“However the precise mechanisms of reduced circulating IgM concentrations leading to susceptibility to bacterial lung infection still remain to be fully elucidated in the context of stroke”*. It will also be important to define the precise adrenergic signalling pathway(s) connecting ischaemic brain damage to systemic B cell disruption – we have amended a sentence in the Discussion (p17) to indicate this *“However we do not exclude that noradrenaline could also act indirectly via an intermediate cell to affect B cells after experimental stroke and the precise mechanisms of how brain ischemia results in increased splenic noradrenaline remain to be determined”*.

Reviewer #3 (Remarks to the Author):

The author's responses to reviewer 3 are acceptable.

Response: We thank the reviewer for their previous constructive comments and are pleased the amendments we have made are satisfactory.